# Toward Sustainable Development: Decoupling the High Ecological Footprint from Human Society Development: A Case Study of Hong Kong

**Xiangyun Shi [1],\*, Takanori Matsui [1] , Takashi Machimura [1], Xiaoyu Gan [2] and Ang Hu [2]**

[1] Division of Sustainable Energy and Environmental Engineering, Graduate School of Engineering, Osaka University, Osaka 5650871, Japan; matsui@see.eng.osaka-u.ac.jp (T.M.); mach@see.eng.osaka-u.ac.jp (T.M.)

[2] College of Architecture and Environment, Sichuan University, Chengdu 610065, China; ganxy@scu.edu.cn (X.G.); ang.hu6@scu.edu.cn (A.H.)

\* Correspondence: xiangyun.shi@ge.see.eng.osaka-u.ac.jp

**Abstract:** As a global financial center and one of the world's first-tier cities, Hong Kong is committed to sustainable development and it expects to become the most sustainable city in Asia. With this in mind, this paper evaluates the level of sustainable development in Hong Kong considering the factors of ecological footprint, biocapacity, and the human development index (HDI) from 1995 to 2016, in order to make policy recommendations for transforming Hong Kong into a more sustainable city. Between 1995 and 2016, a period during which the HDI rose, the per capita ecological footprint of Hong Kong increased from 4.842 gha to 6.223 gha. Moreover, fossil energy consumption had a crucial impact on the city's ecological footprint, whereas the biocapacity of Hong Kong declined gradually. By contrast, Singapore, a city-state with an area similar to Hong Kong's, presented the opposite situation—the HDI increased while the ecological footprint decreased. We performed a further comparative analysis and a SWOT analysis of Singapore and Hong Kong to elaborate on how to decouple the large ecological footprint from human society development. Concluding that the focus must be on energy consumption, reduction of the human activities' negative impacts on marine environment, citizens and government, we provide policy suggestions for transforming toward a "high HDI and low footprint" sustainable development society in Hong Kong.

**Keywords:** sustainable development; decoupling; ecological footprint; biocapacity; human development index; Hong Kong; Singapore

## 1. Introduction

*1.1. Sustainability Indicators: The Role of the Human Development Index and Ecological Footprint*

In 1972, the United Nations Conference on the Human Environment put forward the *Declaration on the Human Environment*, an appeal to tread cautiously to avoid environmental destruction and a call on all humanity to protect the environment for themselves and future generations. This laid the foundation for sustainable development theory [1]. In 1987, the sustainable development model was formally proposed in *Our Common Future,* a report by the World Commission on Environment and Development. Sustainability science was first illustrated in the journal Science by Kates et al. [2] in 2001, which accelerated the development of this field. In 2015, the United Nations put forward 17 Sustainable Development Goals (SDGs) as the heart of the 2030 Agenda for Sustainable Development. These were to provide a shared blueprint for peace and prosperity for human society and our planet, now and into the future. At present, sustainable development has become the North Star of the international community [3].

Over the last two decades, there has been a prominent increase in the number of sustainability indicators and methods for tracking progress toward sustainability [4]. The ecological footprint approach has demonstrated good universality when considering different spatial scales—whether global, national or regional [5,6]. It can be used to assess both current ecological supply and demand and historical trends to provide a basis for setting goals, identifying options for action and tracking progress toward the stated goal. Furthermore, compared with other indicators, ecological footprint accounting allows for three unique approaches: (I) consumption (ecological footprint) can be compared to a biophysical budget limitation (biocapacity); (II) data can be aggregated to a single comparable unit of biocapacity (gha); and (III) time series of flows can be provided [7]. Moreover, the ecological footprint correlates with sustainable development and both take into account the following factors: (i) the increase in human consumption and its consequences; (ii) the key resources for sustainable development, i.e., the biological production of land and oceans; (iii) the distribution of available resources; and (iv) the impact of trade on sustainable development and the redistribution of regional resources under environmental pressures [8].

However, the ecological footprint measurement does not incorporate human dimensions [9] that are necessary for indicating how societies should develop. The Human Development Index (HDI) was created as an overarching and composite index to evaluate the well-being of human societies. That is, human development is about expanding the richness of human life, rather than focusing on the richness of the economy only. As a summary measure, HDI evaluates long-term progress in three basic dimensions: a long and healthy life, access to knowledge, and a decent standard of living [10]. These three dimensions correspond to three of the SDGs: Goal 3 (good health and well-being), Goal 4 (quality education), and Goal 8 (decent work and economic growth). The HDI has become widely accepted as a useful metric in the sustainability field [11].

With respect to the application of HDI in China, before 2000, economic development was China's major target, so social policies were poorly adopted. Education and health received minor attention during much of the human development of China. However, since the start of the millennium, social policies have received more attention [12], although the gap between education, health and economic growth is still significant and gradually increasing.

Ecological footprint research in China can be classified into two categories [13]: (I) adopting the ecological footprint method to synthetically measure the demand of natural resources on national and provincial scales, and (II) describing the ecological footprint of a particular production/consumption, such as tourism, transportation, water resources, etc. Although considerable ecological footprint-related research has been conducted in China, some deficiencies still exist: (i) research tends to focus on national and provincial areas and tends to neglect small scales, such as urban and rural areas [14]; (ii) dynamic research on temporal series is limited and is mainly focused on Midwest regions [15]; and (iii) ecological footprint research in China tends to neglect the Chinese special administrative regions, such as Hong Kong and Macau, due to the limitation of data sources, [16–19], which indicates a research gap in these regions.

### 1.2. Current Challenge of an Asian Megacity—Hong Kong

As sustainable development has been promoted and implemented by the international community and domestic government, it has also been presented in the future strategy of Hong Kong. As an important guideline for planning, the Hong Kong 2030+ [20] pioneers a vision wherein Hong Kong will become an international metropolis of Asia, which advocates sustainable development to meet the current and future requirements of society, the environment and economic growth.

Nevertheless, there is an obvious gap between the current situation and attainment of the SDGs. Environmental protection and resource conservation in Hong Kong is lacking [21]. For instance, per capita, seafood consumption in Hong Kong was ranked second in Asia, and Hong Kong handled about 50% of the global shark fin trade [22], which increased the burden on the global scale. Hong Kong also lacks incentives to improve its energy efficiency and develop renewable energy. For the past decade,

renewable energy accounted for only 0.1% of the primary energy used to generate electricity [23]. Moreover, the rapid growth of the population and economy of Hong Kong also put a heavy burden on land supply. For instance, about 1200 ha extra land area is required to meet the needs of housing (200 ha), economic uses (300 ha) and public spaces (700 ha, including government, institution, community and transport facilities) [20], aggravating the tension in the supply and demand relationship between human development and the environment. On the road to sustainable development, climate change, an aging population and air pollution also present further challenges [24].

### 1.3. Purpose of this Paper

In the context described above, we take Hong Kong as the research object, studying sustainable development issues from the urban perspective, corresponding to Sustainable Development Goal 11, "sustainable cities and communities". The cross-cutting nature of urban issues also impacts other SDGs, such as SDGs 3, 4, 8, 12, 14, and 15 [25]. Therefore, HDI, the ecological footprint, and biocapacity are combined in this paper. HDI evaluates the human well-being that relates to a healthy life (Goal 3), education (Goal 4), and a decent standard of living (Goal 8); the ecological footprint measures the natural resources consumed by the urban population (Goal 12, 14, and 15). All above assessments are analyzed in the long-term, namely from 1995 to 2016. Thereby, the characteristics and dynamic changes of Hong Kong development can be observed deeply. Subsequently, a SWOT analysis is performed and Hong Kong is compared with Singapore, a more sustainable city in Asia, in order to elaborate how human society development can be decoupled from a large ecological footprint, and to share recommendations for a sustainable transformation.

## 2. Materials and Methods

### 2.1. Study Area

Hong Kong is located in the southeast of China and consists of Hong Kong Island, Lantau Island, Kowloon and the New Territories (including 262 outlying islands). Due to its geographical location, half of the world's population can be reached within 5 h of flying time from Hong Kong (Figure 1). The total population in 2016 was 7.34 million, the average population growth rate from 1995 to 2016 was 1.08% per annum, and by 2043, the population will increase to 8.22 million [20]. The total land area is approximately 1110 square kilometers and the sea area is about 1649 square kilometers. At present, less than 25% of Hong Kong's land has been developed, and parks and nature reserves account for approximately 40%. Hong Kong's economy is dominated by the service industry. Trading and logistics, financial services, producer and professional services and tourism are the four pillar industries of Hong Kong. It also enjoys a worldwide reputation for its financial center.

### 2.2. Data Sources

To ensure the accuracy and reliability of the research, the data in this paper were mainly derived from the Hong Kong Annual Digest of Statistics (2001–2018) [27], the Hong Kong Energy Statistics [28], the Agriculture, the Fisheries and Conservation Department Report of Hong Kong [29], the Food and Agriculture Organization of the United Nations (FAO) [30], the Yearbook of Statistics Singapore [31] and the United Nations Development Programme [10].

### 2.3. Ecological Footprint

The ecological footprint calculates the combined demand for ecological resources and energy, focusing on six main categories of biologically productive land that are required by human activities: arable land, grazing land, forest land, water area, fossil land and built-up land [32]. However, the ecological footprint calculated in a case can vary depending on the accuracy of statistical data, the scope of analysis, and the use of different equivalence factors and yield factors [33]. In this

paper, the calculation of ecological footprint consists of two parts: the biological accounts and the energy accounts, as shown in Table 1.

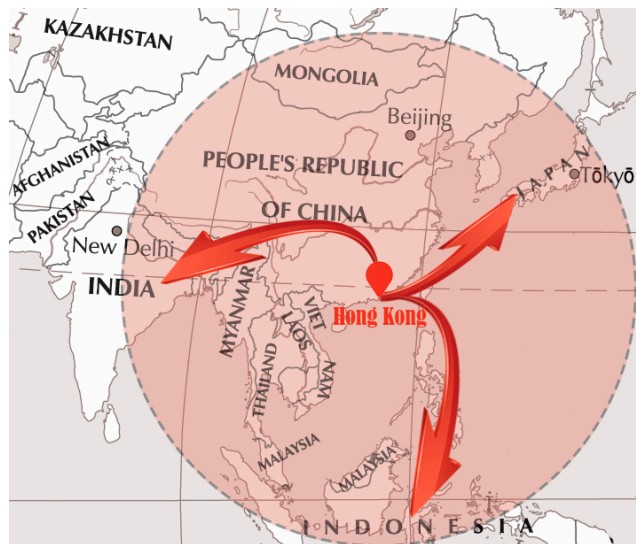

**Figure 1.** Five hours' flying time circle of Hong Kong. (Our own elaboration from Hong Kong 2030+ [20] and Standard Map services [26]).

**Table 1.** Ecological footprint accounts and subjects of Hong Kong.

| Accounts | Subjects | Land Types | Equivalence Factors | Yield Factors |
|---|---|---|---|---|
| Biological accounts | rice, wheat, cereals, vegetables, tea, sugar and honey, pigs, other poultry, chickens, meat and meat preparations | arable land | 1.96 | 1.65 |
| | fruit, timber, coffee, cocoa | forest land | 0.98 | 1.03 |
| | cattle, sheep, dairy products | grazing land | 0.82 | 2.71 |
| | fish and fishery products, crustacean, mollusks. | water area | 0.64 | 2.71 |
| | residential, commercial, industrial, open area, transportation | built-up land | 1.96 | 1.65 |
| Energy accounts | Coke oven gas, kerosene, gasoline, diesel oil, fuel oil, LPG, natural gas, coal, electricity | fossil land | 0.98 | 0.00 |

Note: Equivalence and yield factors for Hong Kong were derived from Liu and Li [34]; Liu, Li and Xie [35].

The biological accounts are mainly related to agricultural products, forest products, livestock products, and aquatic products. As for the energy accounts, according to the *General Principles for Calculation of Comprehensive Energy Consumption* (GB/T2589-2008) in China, the low calorific value generated by 1 kg of fossil fuel is taken as a standard to convert the energy consumption into fossil energy land and construction land, as shown in Table 2.

**Table 2.** The average low heat value of energy and global average specific energy footprint.

| Types | Average Low Heat Value GJ/t | Specific Energy Footprint Global Average in GJ/ha per Year | Land Types |
|---|---|---|---|
| Kerosene | 43.070 | 71 | fossil energy land |
| Gasoline | 43.070 | 93 | fossil energy land |
| Diesel oil | 42.652 | 93 | fossil energy land |
| Coal | 20.908 | 55 | fossil energy land |
| Fuel oil | 41.816 | 71 | fossil energy land |
| LPG | 50.200 | 71 | fossil energy land |
| Coke oven gas | 17.981 | 93 | fossil energy land |
| Natural gas | 35544 [①] | 93 | fossil energy land |
| Electricity | 3600 [②] | 71 | fossil energy land |

Note: [①] The unit is kJ/m$^3$ [②] The unit is kJ/kW·h.

In addition, due to the vast territory of China, the land productivity of each province is different. Liu and Li [34] and Liu, Li, and Xie [35] used a net primary productivity approach to calculate the equivalence factors and yield factors for each province. We adopted those factors that correspond to Hong Kong to make the results more reliable (Table 1).

The basic equation of the ecological footprint EF (gha) [36] is

$$EF = \sum_i \frac{P_i}{Y_{w,i}} \cdot EQF_i \tag{1}$$

The per capita ecological footprint ef (gha/cap) is

$$ef = \frac{EF}{N} = \sum_i \frac{P_i}{N \cdot Y_{w,i}} \cdot EQF_i \tag{2}$$

where P (kg) is the amount of each primary product i that is harvested; N is the population; $Y_{W,i}$ (kg ha$^{-1}$) is the average world yield for commodity i; and $EQF_i$ is the equivalence factor for the land use type producing products i. The detailed calculation methods of $Y_{W,i}$ and $EQF_i$ are explained in reference [34,35].

This paper uses a consumer-based approach, which measures the biocapacity demanded by the final consumption of the region in question. The consumer-based method has become the most widely used calculation [37], especially for regions that rely heavily on imported products due to poverty in natural resources, such as Hong Kong.

For each land use type, the ecological footprint of consumption ($EF_C$) [38] is thus calculated as

$$EF_C = EF_P + EF_I - EF_E \tag{3}$$

where $EF_C$ (gha) is the ecological footprint of consumption, which indicates the consumption of biocapacity by inhabitants. $EF_P$ (gha) is the ecological footprint of production, which indicates the consumption of biocapacity resulting from production processes within a given geographic area. $EF_I$ (gha) and $EF_E$ (gha) are the ecological footprint of imports and exports, respectively, and indicate the use of biocapacity within international trade.

According to the subjects in Table 1, after transferring the corresponding values into Equation (2), and calculating the $EF_P$, $EF_I$, and $EF_E$ of the six land types, Equation (3) can be used to calculate the $EF_C$.

### 2.4. Biocapacity

Biocapacity (BC) is a measure of the amount of biologically productive land and sea area available to provide the ecosystem services that humanity consumes—as our ecological budget or nature's regenerative capacity, it represents the total area of biologically productive land that the region can provide to humans [36]. The equation is

$$bc = \sum_i \frac{A_i}{N} \cdot YF_i \cdot EQF_i \tag{4}$$

where bc (gha/cap) is the per capita biocapacity; $A_i$ (ha) is the available area of a given land use type; N is the population; and $YF_i$ and $EQF_i$ are the yield factors and equivalence factors, respectively, for the land use type.

### 2.5. Ecological Reserve/Deficit

Ecological reserve and ecological deficit are based on the calculation of the regional ecological footprint and biocapacity to ascertain whether the demands of society exceed the regional biocapacity. Using these variables, a determination can be made about whether development in the region is sustainable [39]. The equation is

$$\begin{cases} ed = ef - bc & (ef > bc) \\ er = bc - ef & (ef < bc) \end{cases} \tag{5}$$

where ed (gha) is the total ecological deficit; er (gha) is the total ecological reserve; bc (gha) is the total biocapacity; and ef (gha) is the total ecological footprint. If bc − ef > 0, then there is an ecological reserve (er), indicating that the demands of humans in the region are within its biocapacity; if bc − ef < 0, then there is an ecological deficit (ed), indicating that the demands exceed the biocapacity.

### 2.6. Human Development Index

As an overarching index, HDI was created for assessing three key dimensions of human development: a long and healthy life, acquisition of knowledge and a decent standard of living. An overview of the calculation of HDI is shown in Figure 2.

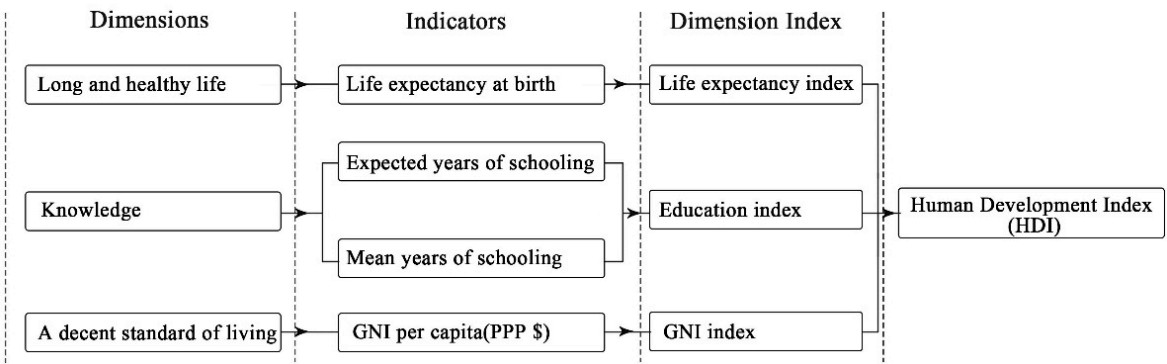

**Figure 2.** The calculation of human development index (HDI). (Our own elaboration from United Nations Development Programme [10]).

The calculation process consists of (1) identifying the minimum and maximum values of life expectancy at birth, expected years of schooling, mean years of schooling and GNI, respectively. Afterwards, it involves transforming these different unit indicators into indices between 0 and 1 by applying Equation (6):

$$\text{Dimension index} = \frac{\text{actual value} - \text{minimum value}}{\text{maximum value} - \text{minimum value}} \tag{6}$$

(2) Finally, all indices are aggregated by Equation (7)

$$\text{HDI} = \left(I_{\text{health}} \cdot I_{\text{education}} \cdot I_{\text{income}}\right)^{\frac{1}{3}} \tag{7}$$

where $I_{\text{health}}$, $I_{\text{education}}$, and $I_{\text{income}}$ are the results calculated by Equation (6).

## 3. Results and Analysis

### 3.1. Results of Ecological Footprint

Using Equations (1)–(3), the six categories of biologically productive lands of Hong Kong were calculated from 1995 to 2016, respectively. Figure 3 (for details, please see Table S1) illustrates that the per capita ecological footprint of Hong Kong increased from 4.842 gha/cap (1995) to 6.223 gha/cap (2016), with an average annual growth rate of 1.3%. The ecological footprint of arable land, fossil land, and water area made up the main consumptions of Hong Kong.

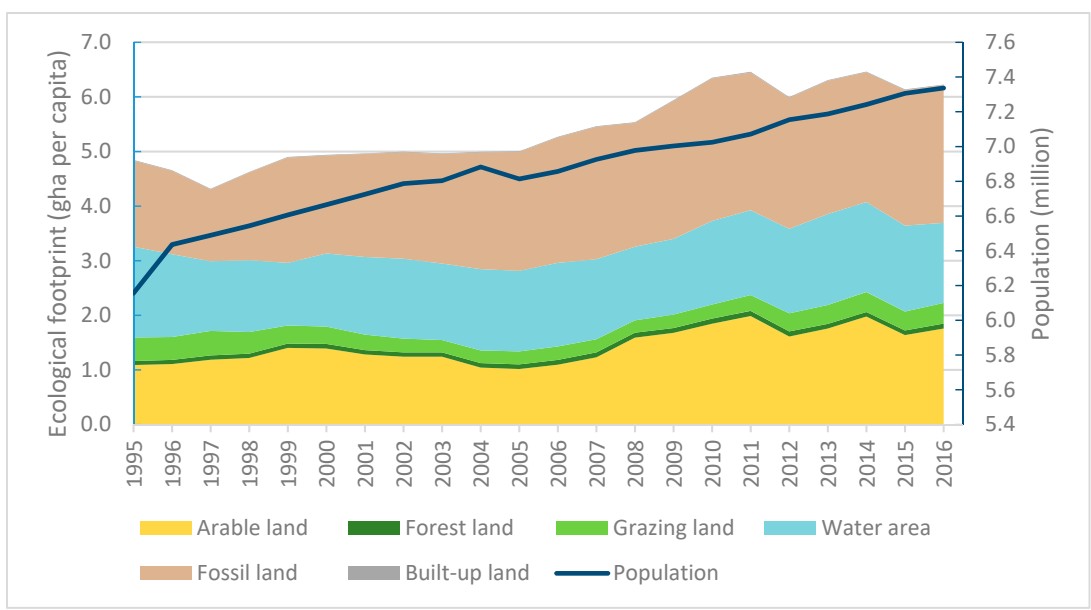

**Figure 3.** Hong Kong's population and per capita ecological footprint by land category during 1995–2016.

With regard to the biological accounts, according to statistical data from Agriculture, Fisheries and Conservation Department, Food and Health Bureau [40], in 2016, the most common types of fresh food consumed each day (ranked from most to least consumed) were vegetables, fruits, pigs, salt-water fish and eggs. More than 90% of this food was supplied by mainland China. In order to maintain stable supply chains, frozen food was also imported from Brazil, Norway, Philippines and Thailand. It can be seen that the ecological footprint of Hong Kong does not just cover a regional scale, but extends to a global scale.

As for the energy accounts, fossil energy consumption accounted for the largest proportion of growth (Figure 3). It rose from 1.578 gha/cap to 2.518 gha/cap between 1995 and 2016, with an average proportion of 39%. Figure 4 (for details: Table S2) shows the components of Hong Kong's energy ecological footprint. Fuel oil, coal, electricity, and kerosene were the major types of energy consumption. Although the proportion of coal consumption declined slightly over time, it was still the largest type of consumed energy. Coal was the primary source of energy for generating electricity [28].

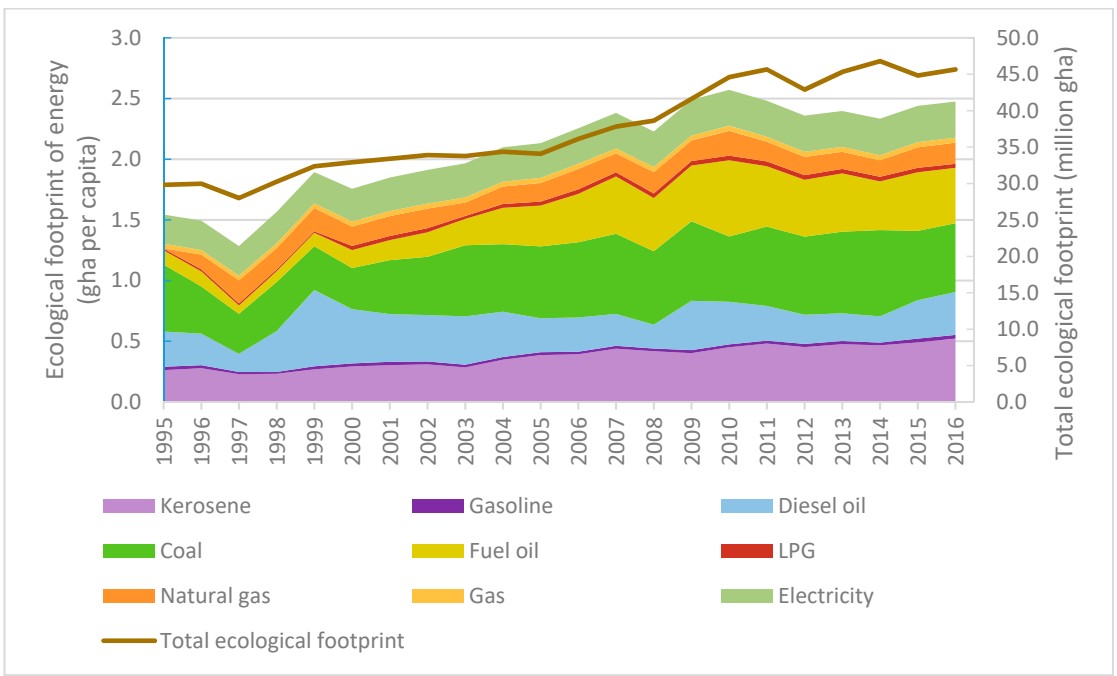

**Figure 4.** Hong Kong's total ecological footprint and per capita ecological footprint of energy. Each area shows the major components of energy consumption each year during 1995–2016. There are two main sources of electricity supply in Hong Kong: the imports of electricity from the mainland of China and the electricity generated at local plants. Coal is the main source of energy of local electric plants.

*3.2. Results of Biocapacity*

As shown in Figure 5, from 1995 to 2016, biocapacity declined over time (for details: Table S3). The biocapacity of grazing land decreased significantly, from 0.0192 gha/cap to 0.0058 gha/cap, followed by arable land, which declined from 0.0042 gha/cap to 0.0030 gha/cap, while the biocapacities of forest land and built-up land increased. The biocapacity of water land remained constant.

Arable land represents the agriculture area in Hong Kong, grazing land represents the grassland, forest land includes woodland, shrubland, mangrove and built-up land includes residential area, commercial area, industrial area, open space, the land for transportation, etc. The water area mainly represents reservoirs, streams and nullahs.

In 1995, the agriculture area and built-up land area were 9100 ha and 17,200 ha, respectively. The vegetation area, including forest, shrub, grassland and wetland, was 78,300 ha. However, as the land became more urbanized, by 2016, the agriculture area, vegetation area and built-up land were 6800 ha, 73,600 ha, and 27,000 ha, respectively. In order to meet the needs of urban expansion, the agriculture and vegetation areas were shrinking, and the abandoned farmland and grassland contributed to the built-up land.

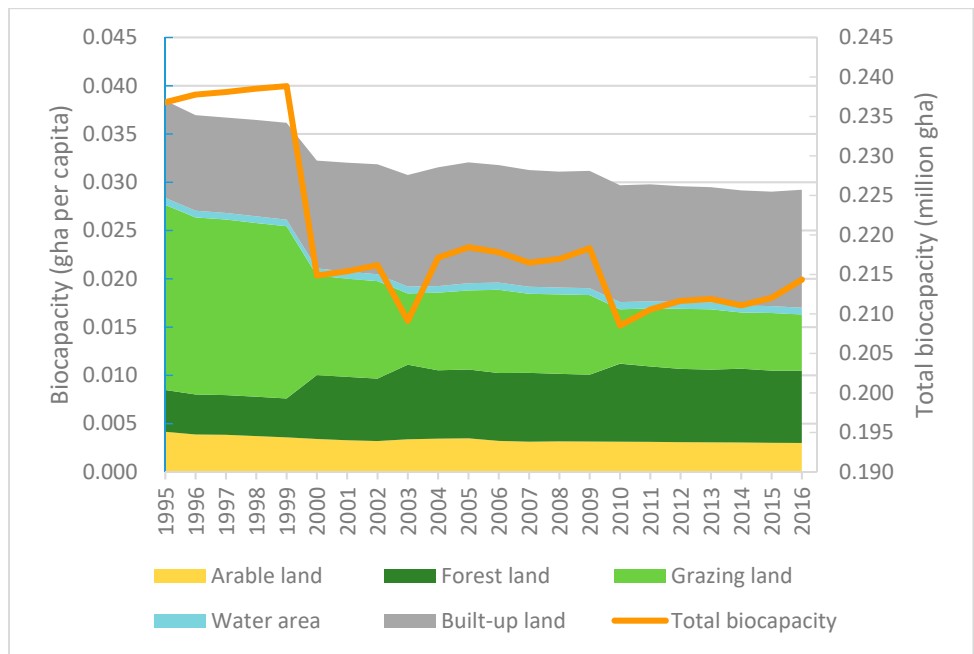

**Figure 5.** Hong Kong's total biocapacity and per capita biocapacity by land category during 1995–2016.

### 3.3. Results of Ecological Deficit

According to the Equation (5), the ecological deficit results can be calculated based on ecological footprint and biocapacity, as shown in Figure 6. Obviously, consumption by Hong Kong's population exceeded the local biocapacity, and the trend continued each year. Compared with 1995, the ecological deficit increased by approximately 1.3 times by 2016. As a natural resource poverty city, Hong Kong relies heavily on the ecosystem outside its own boundaries [33]. Further, Hong Kong's fisheries, which were historically unrestricted and very exploitative, have already deteriorated the marine ecosystem around Hong Kong [41]. In addition to overfishing, the negative effects resulting from reclamation and waste discharge are also considerable. With the increasing consumption of the growing population, these human activities are eroding the ecosystem, resulting in a weaker biocapacity and a more aggravated ecological deficit.

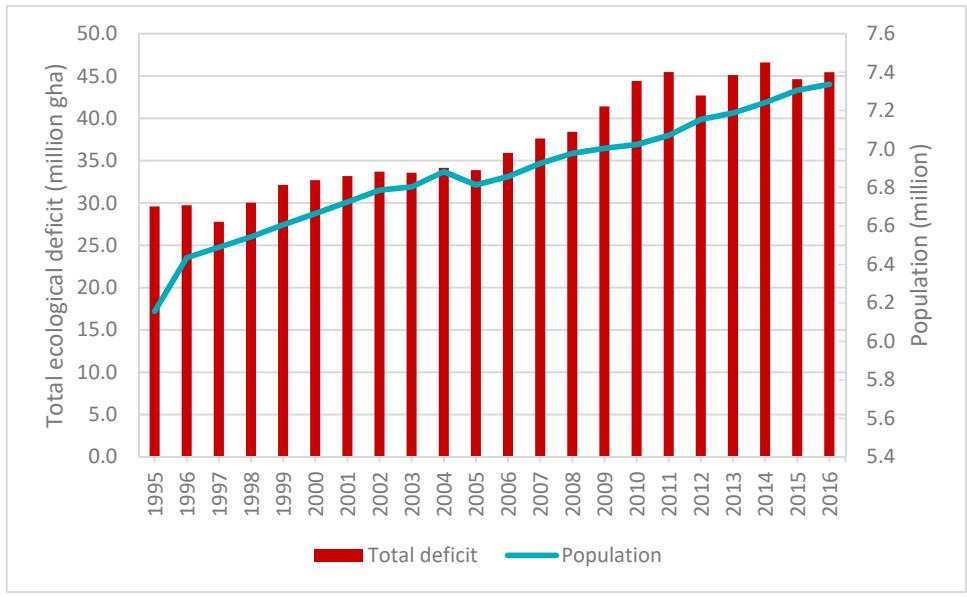

**Figure 6.** Hong Kong's population and total ecological deficit during 1995–2016.

### 3.4. Human Development Index of Hong Kong

The HDI and its component indicators, namely life expectancy index, education index, and GNI per capita of Hong Kong are shown in Figure 7. In 1995, the HDI of Hong Kong was 0.808 and followed a trend of increasing over time, so that by 2010, Hong Kong's HDI exceeded 0.9, which indicates that Hong Kong had achieved a high level of human well-being. Moreover, all three of these component indicators showed an upward trend over 22 years. According to the World Bank classification, high-income economies are those with a GNI per capita more than $12,746 [42]. In 1995, Hong Kong's GNI ($32,678, Figure 7) was already higher than this value, and by 2016, the GNI of Hong Kong was $55,809, with an average annual growth rate of 2.71% from 1995 to 2016, which implies that for the people of Hong Kong, both incomes and quality of life improved over time. The life expectancy index was consistently at the highest level among these indicators. The life expectancy at birth rose from 79.0 years (1995) to 84.0 years (2016), an above-average value for most countries in East Asia and the Pacific, such as China (76.3 years, 2016), Singapore (83.0 years, 2016), and Japan (83.8 years, 2016) [43], reflecting the relatively good health of the people of Hong Kong.

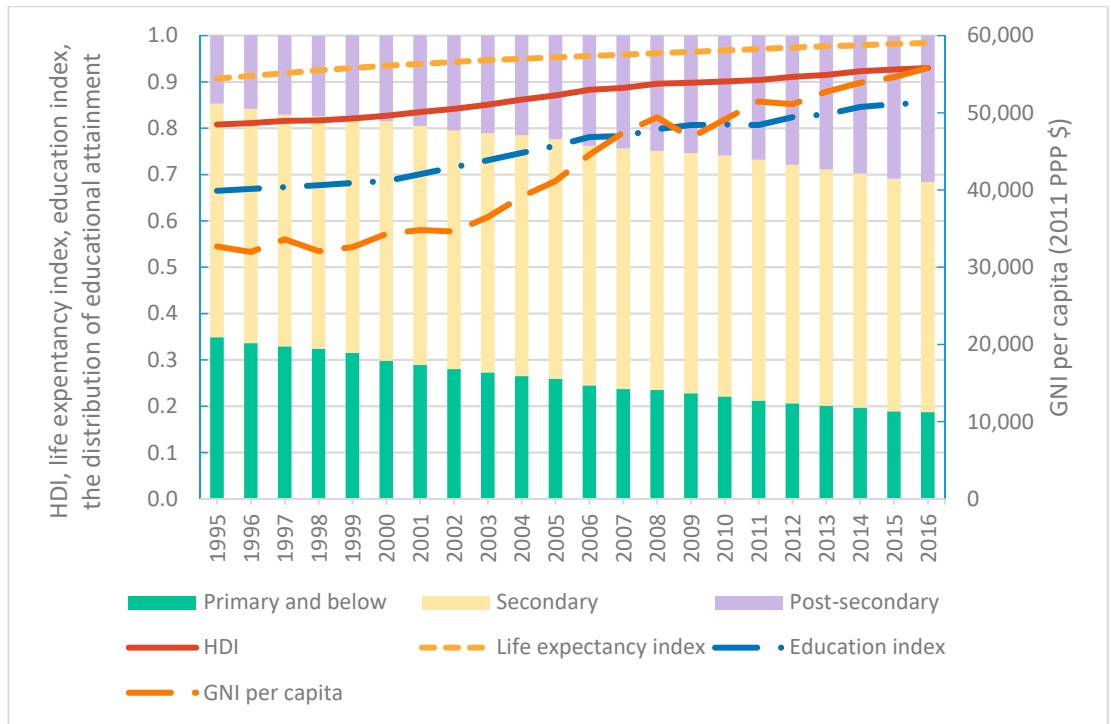

**Figure 7.** Life expectancy index, education index, GNI per capita, HDI of Hong Kong and the distribution of educational attainment of people aged 15 and over. The primary and below group includes people with no schooling, or kindergarten or a primary level of education. The secondary group includes people with lower secondary and upper secondary education. The post-secondary includes people with technical schooling, or non-degree or degree coursework. (This figure was drawn by our own elaboration. Data sources: Life expectancy index, education index, GNI per capita, and HDI data are sourced from the UNDP-Human development data [43]. The distribution of educational attainment of population data is sourced from the Census and Statistics Department of Hong Kong [44]).

The education index was lower than 0.7 between 1995 and 2000. However, from 2001 to 2008, it increased meaningfully to exceed 0.8, which reflected a higher knowledge level in Hong Kong. To illustrate this change further, as shown in Figure 7, the distribution of educational attainment of population, it was not just that the expected years of schooling increased each year (from 13.4 (1995) to 16.3 (2016)), but also the structure of educational attainment that was optimized. Moreover, since 2010, the category of those whose education ended at the secondary level (including technical, non-degree

and degree courses) also began to show a downtrend, indicating a trend of people obtaining higher levels of education, creating a healthier and more sustainable society in Hong Kong.

Above all, when sustainability is viewed through the lens of human development, the HDI of Hong Kong indicates a high level of sustainability, and in 2017, it ranked seventh in the world.

## 4. Discussion and Implications

### 4.1. The State of Sustainable Development in Hong Kong

Sustainable development of human society relates not only to the improvements of humanity, but also the sustainability of the environment. Therefore, the ecological footprint and biocapacity were adopted to measure human consumptions and natural supplies. Figure 7 shows that the economic growth and human improvements of Hong Kong progressed significantly from 1995 to 2016. But this also caused an obvious increase in the size of the ecological footprint and a decline in biocapacity, which resulted in a serious ecological deficit (Figure 6). This reveals that although both economy and human well-being improved in Hong Kong, in terms of environmental considerations, development in Hong Kong is still following an unsustainable development trajectory.

Naveh [45] asserted that the human urban ecosystem was primarily driven by fossil fuels, as is the case in Hong Kong (Figure 3), and environmental pollution and greenhouse gas emission will be aggravated by the heavy dependence on fossil fuel energy. In turn, the effects of climate change, such as a rising temperature could also impact energy consumption [46].

Second, it is notable that the ecological footprint of the water area was also very large (Figure 3), even larger than the fossil fuel footprint before 1998. As the productivity of local waters is very limited, Hong Kong's seafood consumption mainly relies on imports from other parts of the world. The overconsumption has had a significant negative impact not only on the local marine ecosystem but also on the Indo-Pacific [47]. Unfortunately, such problems have not been mitigated in recent years but have tended to escalate. By 2016, the local fishery resources dropped by 27% [48]. At present, only 1.5% of all marine areas are designated as being protected by Hong Kong, and there are other threats to the biocapacity of the water area: (I) the lack of an explicit coverage area, goals, or a time schedule for the establishment of a future marine protection zone; (II) the lack of specific and effective management for existing protection zones; and further, (III) the failure of the government and experts to effectively organize positive discussion for the protection site selection, and failure of fishermen to timely participate [49].

In brief, although sustainability viewed through the lens of human development was maintained at a high level in Hong Kong, the large ecological footprint and environmental issues prevent Hong Kong from approaching Goal 12 (sustainable consumption), 14 (sustainably use the marine resources) and 15 (sustainable use of terrestrial ecosystems). To prevent the negative impacts mentioned above and move toward a more sustainable development future, the large ecological footprint should be decoupled from human society development.

### 4.2. Comparison of Hong Kong with the Best Sustainable Practice City-States—Singapore

As Asian city-states, Singapore and Hong Kong have many similarities. Both were colonized by the British in the nineteenth century and created by immigrant populations from China, and both have economies that grew out of their status as entrepots [50]. Both face the problems of limited land, high urban density, and poverty of natural resources.

In the 1960s, Singapore started its journey toward sustainability. Thereafter, a series of policies and movements were implemented, such as the Keep Singapore Clean Campaign in 1968 and the Clean Air Act in 1971, which transformed Singapore into a more sustainable city [51]. However, the development of Hong Kong shows an obviously different trend. For instance, although the HDI of Hong Kong and Singapore both reached 0.93 in 2016 (Figure 8), the size of their ecological footprints has diverged in

recent years. In order to explore how a human society can advance without an increased ecological footprint, we compare Hong Kong with Singapore.

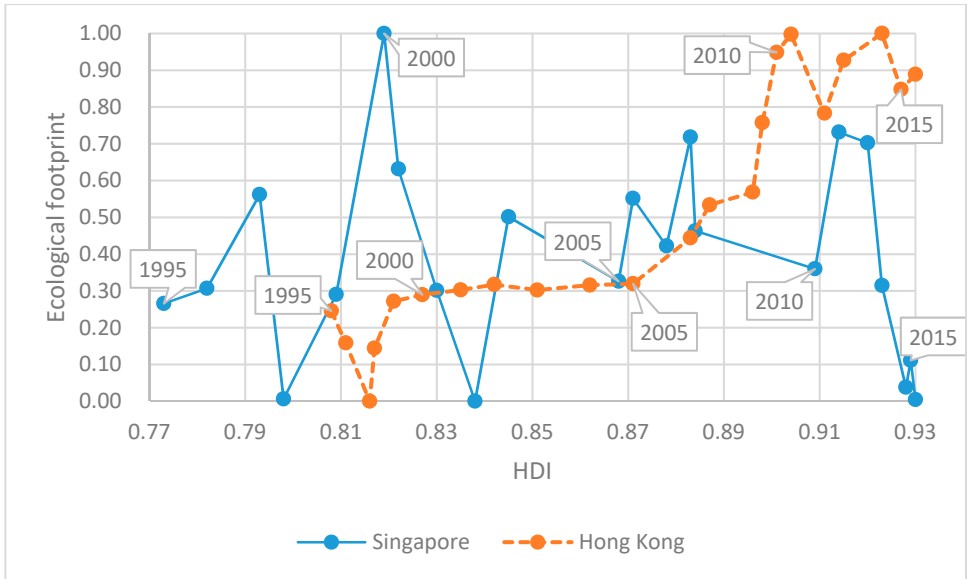

**Figure 8.** The normalization of HDI and the ecological footprints of Hong Kong and Singapore. Because the ecological footprint results of Singapore and Hong Kong are from different data sources, they cannot be compared directly. Therefore, to focus on the comparison of trends instead of the specific numerical value, the rescaling (min-max normalization) method was used to present the vertical axis. This method does not change the distribution characteristics of data. The general equation for a min-max of [0, 1] is given as: $X_{norm} = \frac{X - X_{min}}{X_{max} - X_{min}}$ where X is an original value, $X_{norm}$ is the normalized value, $X_{max}$ and $X_{min}$ are the maximum and minimum of the original dataset. (The HDI values of Hong Kong and Singapore are our own elaboration from UNDP- Human development data [43])

Singapore is an island city-state in Southeast Asia with only 719.2 km$^2$ land area and a population of about 5.6 million (2016). Although it is about half the size of Hong Kong, its natural resource consumption rate is quite high. Singapore had one of Asia-Pacific's largest ecological footprints per capita [52]. Similar to Hong Kong, the consumption of the water area resource was also very large, as an average of more than 100,000 tons of seafood is consumed each year in Singapore, making it one of the biggest seafood consumers in the Asia-Pacific region [53]. In addition, the fossil fuel footprint historically took up the largest proportion of the total ecological footprint (Figure 9, Table S4)–its average value over the past 22 years is 5.36 gha/cap.

However, it is notable that the size of Singapore's ecological footprint has decreased continuously since 2011, while its HDI has continued to rise (Figure 8). By contrast, in this time, the size of Hong Kong's ecological footprint has continued to increase.

To decouple an increased ecological footprint from improvements in human society, Singapore decreased its fossil fuel footprint and followed by reducing the forest land footprint. Market liberalization of Singapore has led to rapid replacement of oil-fired steam plants with gas-fired combined-cycle gas turbines, which has lowered the carbon intensity [55]. Besides, compared with the pipeline natural gas that imported from Indonesia and Malaysia, the liquefied natural gas, a form of natural gas that can be more easily transported on a global scale, has started to be imported in 2013, to diversity and secure energy source for Singapore [56]. Furthermore, solar photovoltaic (PV) systems have been actively deployed since 2010 [57]. Beyond 2020, the adoption of solar power will be further raised to 1 GigaWatt peak (GWp). Undoubtedly, this will help Singapore to achieve its climate change pledge of reducing the emissions intensity by 36% from 2005 levels by 2030 [58].

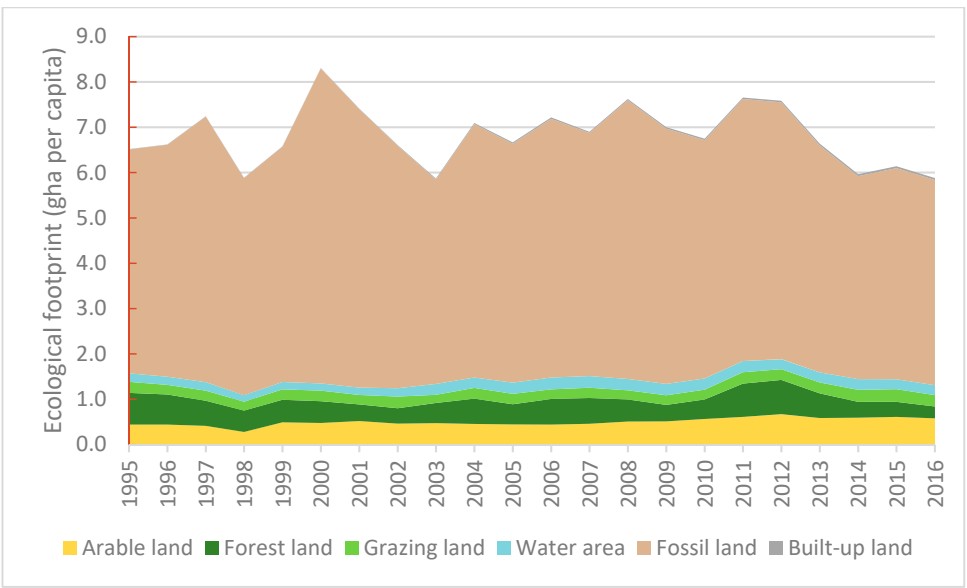

**Figure 9.** Singapore's per capita ecological footprint by land category during 1995–2016. (Our own elaboration from Global Footprint Network [54]).

On the other hand, because Singapore has no wind, hydro or geothermal resources, and the land for deploying solar power is also limited, the government must rely on more innovative, resilient and sustainable energy strategies:

(I).　Transportation. One of the key elements in managing environmental footprint [59] and urban sustainable development is sustainable transport [60]. In Singapore, it has been achieved by enhancing public transport, improving resource efficiency, and reducing carbon emissions [59], such as the use of electric vehicles was assessed on a national scale, and it was determined that their efficiency was higher than that of gas vehicles [61].

(II).　Electricity. The government is optimizing smart metering technology to reduce the cost of electricity. These automation devices could help people cut wasteful or unintentional usage, and potentially shift usage patterns to off-peak periods when the electricity price is lower. Moreover, this technology has been introduced as a part of the Intelligent Energy System project, which uses a smart grid to better manage electricity [61].

(III).　Raising the awareness of energy conservation. For instance, a new mobile app was created to help households compare their electricity, water, and gas consumption with neighbors, to enable consumers to use energy efficiently, and to potentially lower their utility bills and carbon footprint [62].

From 1995 to 2016, the biocapacity trends of Hong Kong and Singapore were obviously opposite. although total land areas increased in both locations (Figure 10). With regard to the biocapacity of Singapore, the biocapacity of water area and forest land decreased gradually, whereas, the striking growth of built-up land caused the total biocapacity to rise significantly (Figure 11, Table S5). This mainly resulted from long-term sea reclamation. The land area of Singapore increased from 581.5 km² to 719.2 km² between 1965 and 2016, namely, land area increased by 23.6%. Before 2000, the reclamation area grew slowly, increasing by 5.98 km² from 1990 to 2000. However, the reclamation rate rose dramatically to 1185% between 2000 and 2010 [63]. This is why the biocapacity of Singapore has increased rapidly since 2000.

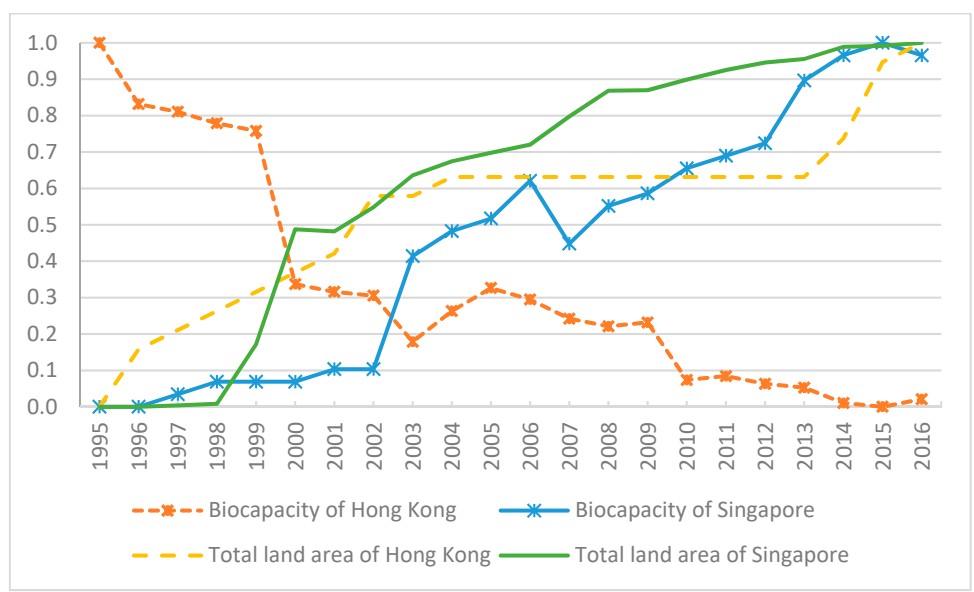

**Figure 10.** The normalization of biocapacity and total land area of Hong Kong and Singapore.

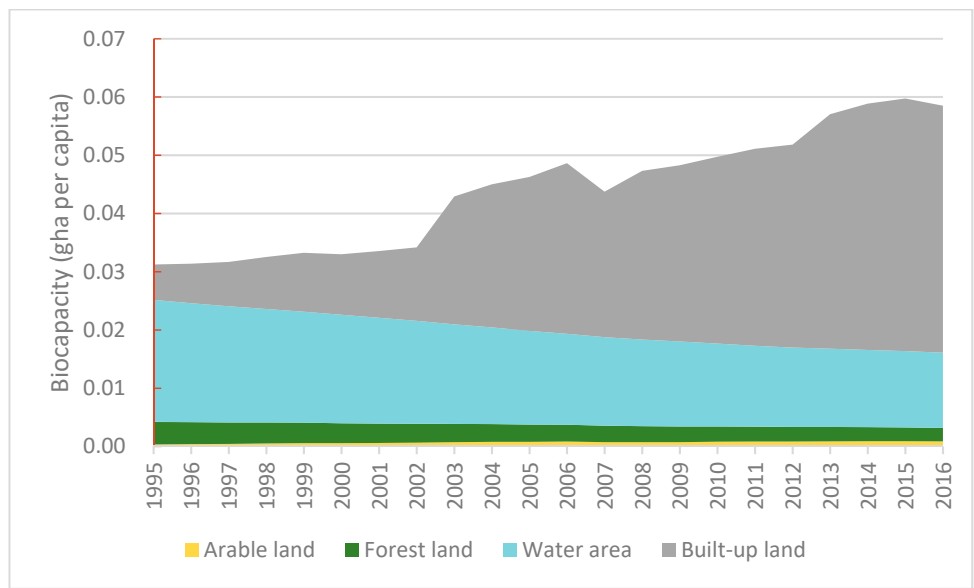

**Figure 11.** Singapore's per capita biocapacity by land category during 1995–2016. (Our own elaboration from Global Footprint Network [54]).

Although reclamation projects were also implemented in Hong Kong, the reclamation rate was far less than in Singapore. The total land area increased from 1092 km² (1995) to 1111 km² (2016) with an average annual growth rate of 0.08%. In addition, the brownfield and deserted agricultural land in rural areas will also be developed into urban space [20]. Therefore, it can be expected that the biocapacity of arable land and grazing land will shrink further, whereas the biocapacity of built-up land will increase.

The high rate of reclamation in Singapore had come at a significant cost to the marine ecosystem. About 65% of the original reefs had been lost to reclamation since the 1960s [64]. With the increased public awareness of the need for environmental protection, the government realized that the reclamation paradigm needed to change, which led to more positive recent actions. This transformation provides an optimized way to balance the socio-ecological system and make the coastal ecosystem more sustainable where the development is intensive.

### 4.3. SWOT Analysis and Policy Implications for Decoupling Ecological Footprint from the Development of Human Society

In order to provide scientific evidence and tailor-made suggestions for policy based on the comparative analysis above, a SWOT analysis was adopted to summarize the current strengths, weaknesses, future opportunities and threats to Hong Kong and Singapore. SWOT analysis is a tool used for formulating strategic planning and management, helping planners leverage strengths and opportunities to reduce weaknesses and threats [65], as shown in Table 3.

**Table 3.** SWOT analysis of sustainable development of Hong Kong and Singapore.

| SWOT | Hong Kong | Singapore |
|---|---|---|
| Strengths | ●Strategic geographical location.<br>●Global financial hub.<br>●Well-protected terrestrial nature reserves, such as country parks [20].<br>●High level of human development.<br>●Sustainable transport system [66]. | ●Advantageous geographic position.<br>●The central trade hub in Asia.<br>●Effective government.<br>●High level of human development.<br>●Relatively strong in innovation and technology [67].<br>●High vegetation coverage rate.<br>●Sustainable transport system. |
| Weaknesses | ●Limited developable land.<br>●Relatively weak in innovation and technology [24].<br>●Air pollution and municipal solid waste generation and disposal [24].<br>●High resource consumption.<br>●Government deficiencies in efficiency of environmental protection [68,69]. | ●Lack of land resource.<br>●High resource consumption, especially for the water area.<br>●No indigenous energy resources.<br>●Habitat fragmentation and biodiversity loss [70]. |
| Opportunities | ●Optimizing use of developed land and creating new land [20].<br>●Current policies for promoting urban sustainable development, such as creating environmental capacity.<br>●Regional and mainland opportunities [24]. | ●Integrating sustainability directly in policy process.<br>●Singapore is the solar hub for Asia, and clean energy companies are building capabilities, such as wind plants, smart grid [67].<br>●Optimizing space by transforming existing areas into new growth districts [70]. |
| Threats | ●Limited indigenous resources, especially energy resources and the deterioration of water resources [24,71].<br>●Limited land and high urban density.<br>●Climate change [24,70].<br>●Waste Management [24,51]. | |

Note: This table mainly describes the SWOT of the environment and resources-related sustainable development. The "Threats" shows the issues that both Hong Kong and Singapore face.

Sustainable development includes environmental, economic and social sustainability [72]. However, based on the ecological footprint and HDI assessment (see Figures 7 and 8), while the economy and society developed well in Hong Kong, the level of environmental sustainability lagged behind, as shown in the "Weakness" and "Threats" in SWOT analysis (see Table 3). In 1995, Brown [73] claimed that to ensure a sustainable future, people need to carry out an environmental sustainability revolution. To this end, close engagement and support from government are indispensable, and this is where the Hong Kong government falls short (see Table 3). In response, in this section, we put forward some policy suggestions for energy consumption, marine environment protection and changes in the behavior of citizens and government—to decouple a large ecological footprint from the development of human society.

### 4.3.1. Energy Consumption

Because fossil energy consumption took up the largest part of the ecological footprint, there is a great potential to decrease the entire ecological footprint significantly by replacing fossil energy with renewable energy. Singapore is not the only one running on this trajectory—Beijing also presented a positive trend. The energy structure had been adjusted effectively in Beijing, and with an increasing proportion of power generation from renewable sources and the decline of fossil fuel consumption, the ecological footprint decreased by approximately 25.8% from 1997 to 2014 [74]. Hopefully, the ecological footprint can also be significantly reduced in Hong Kong by using renewable energy. We propose these policy suggestions for Hong Kong:

1.  Accelerate the development of the smart metering infrastructure and the smart grid. The electricity accounted for about 50% of energy end-uses between 2004 and 2014 in Hong Kong [75]. Thus, efficiently managing and allocating electricity could significantly contribute to the decline of energy consumption. Although the smart grids have been introduced in Hong Kong already, related research and applications are limited and small in scale [76]. Technical aspects of smart grids, the formulation of a specific policy framework, plans, and implementation of the program should be accelerated.
2.  Deploy/import more renewable energy, such as from mainland China. The government could also simplify market rules and regulations for electricity consumers to make it easier for small consumers to receive payment for injecting renewable energy into the power grid and streamlining the metering requirements for renewable energy owners [57].
3.  Spur research into promising energy technologies and systems-level innovation. The government could formulate a suit of incentive mechanisms to support applied research in smart grid, energy conservation, and storage, etc. For instance, Singapore has awarded more than $100 million in funding to date to support a Research and Development project which aims at addressing Singapore's energy challenges, such as smart grids, solar forecasting and power utilities.

Facilitate the commercialization of energy innovation. For instance, the government can build a bridge to engage company members, industry players, researchers, academia and youth to better communicate needs and opportunities [57].

### 4.3.2. The Protection of the Coastal and Marine Environment

According to the results of Hong Kong's ecological footprint, it is noteworthy that the water area footprint was also relatively large (see Section 3.3, Section 4.1, and Figure 3). However, the marine ecosystem, which provides numerous services to human society, has deteriorated due to the negative impacts of human activities. Therefore, policies and regulations that protect the marine environment need to be formulated and monitored further.

1.  Expand the area of the marine environment protection zone. Although the Hong Kong government has established marine protection zones since 1966, the area covered by protection zones is still quite small to date. WWF has suggested that the area of protection zones should account for at least 10% of Hong Kong's offshore and marine waters before 2020. In addition, Russ et al. [77] noted that the establishment of marine protected areas can also benefit the development of local fishery resources.
2.  Strictly forbid marine litter. Marine litter is a long-standing and prominent problem (see Table 3: "Threats"). Each year, the Hong Kong government cleans about 15,000 tons of marine litter. However, vast quantities of garbage remain [78]. Actually, the source of about 95% of marine litter is local garbage [79], meaning that a significant improvement can be achieved through the efforts of local communities and government.
3.  Do not allow commercial fishing and sea reclamation in marine ecological hotspots (see Section 4.1: point (II) and (III)). Hotspots are the areas featuring exceptional concentrations of endemic species

and species facing the threat of human activities [80]. Protecting these hotspots not only supports the holistic management of marine resources but it can also strengthen the resilience of the marine ecosystem to natural disasters and climate change [81].

4. Apply green infrastructure (GI) to the coastal and reclamation projects. Drawing lessons from the coastal development experience of Singapore (see Section 4.2), protection of the natural environment should be considered while addressing the land requirements for urban expansion. Recently, the GI has received attention and has been widely implemented in many places, such as the US, Canada and Europe. [82]. GI is not only contributing to the enhancement of ecosystem services but it can also support an increasing population's demands for resources [83,84]. It is an environmentally-friendly and economical method for achieving urban sustainable development, resilient communities, and climate change mitigation [82].

### 4.3.3. The Improvement of Citizens' Awareness and Government's Executive Action

To progress toward a sustainable future, the joint efforts of citizens and government are indispensable. Nevertheless, the Hong Kong government lacks efficiency in management and protection of the environment (see Section 4.1: point (I) to (III) and Table 3). In addition, Hong Kong is also characterized by high per capita resource consumption (see Table 3). Therefore, we offer the following suggestions:

1. To citizens: Reduce overconsumption. In light of the International Fashion Consumption Survey report [85], about 68% of Hong Kong people admitted that they consumed far more clothes than they actually need and use. Further, food consumption through waste could be reduced, especially during festivals. For instance, during the Mid-Autumn festival in 2010, about 1.87 million mooncakes were discarded in Hong Kong, which is equivalent to 1200 tons of carbon dioxide emissions [86]. By educating the younger generation through public campaigns and commercial advertisements, overconsumption and the related ecological footprint can be reduced.

2. To government: Separate duties for each government department clearly and strengthen supervision. For instance, as shown in the weaknesses part of the SWOT analysis (Table 3), the vague separation of duties of departments and the simple management mode of garbage disposal in Hong Kong makes environmental action less effective [68,69]. Therefore, the adjustment of relevant government departments is needed so that duties are allocated explicitly and supervision is enhanced. If necessary, enforcement regulations can be formulated according to the situation, to control the behavior of citizens and provide supervisors with support.

## 5. Conclusions

### 5.1. Concluding Remarks

This paper provides an analysis that combines ecological footprint, biocapacity and HDI to assess the sustainable development state of Hong Kong from 1995 to 2016. The results show that the ecological footprint had exceeded the local biocapacity significantly and fossil energy consumption was the biggest contribution. Although Hong Kong's economy, educational level and standard of living have been rising over the past 22 years, the growth behind this was based on the imbalance of environment development, with an increasing ecological deficit year by year. This indicates that there is a significant gap between the current development model and Hong Kong's goal of becoming Asia's most sustainable city [48]. Furthermore, through the comparison of Hong Kong and Singapore, we conclude that there is a great potential to decouple the high ecological footprint from the development of human society, and reach a high HDI and low footprint society by replacing fossil energy with renewable energy. Moreover, an optimum balance value, which is the lowest footprint for maintaining a sustainable human society development, could be explored further to quantify the boundary relation of ecological footprint and human society. Finally, based on the SWOT analysis

and implications from the experience of Singapore, we put forward some policy suggestions for Hong Kong's sustainable transformation. As a world-class metropolis, Hong Kong plays an important role not only in China but also around the world, and we hope this research achievement could assist Hong Kong in achieving Goal 11 of SDGs and making a positive impact on the world.

*5.2. Future Research Prospect—From Ecological Footprint to Nature's Contributions to People (NCP) and SDGs*

As a popular index, the ecological footprint can be calculated and compared for systems of any scale, from an urban scale to a national scale [87]. It was promoted as a policy guide and planning tool for sustainability [88]. However, the ecological footprint for consumption as it relates to the total amount of resources available does not reflect the "production footprint," or differences in intensity of resource use [89]. It has also been criticized as an oversimplification [90]. For instance, in the case of energy consumption, the ecological footprint analysis does not tell the whole story with respect to the environmental impacts of a person or entity [91]. Therefore, we need a more comprehensive and innovative method. In 2018, the Nature's Contributions to People (NCP) framework was put forward by Díaz et al. [92]. The aim is to come up with products that are more likely to be incorporated into policy and practice. NCP considers both positive and negative contributions to people's quality of life. As a more inclusive approach, NCP will improve the relevance and value of expert evidence about nature in achieving the SDGs [93].

**Supplementary Materials:** The following are available online at http://www.mdpi.com/2071-1050/12/10/4177/s1, Table S1, 1995–2016 Hong Kong's per capita ecological footprint by land categories, Table S2: 1995–2016 Hong Kong's ecological footprint of energy, Table S3: 1995–2016 Hong Kong's per capita biocapacity by land categories, Table S4: 1995–2016 Singapore's per capita ecological footprint by land categories, Table S5: 1995–2016 Singapore's per capita biocapacity by land categories.

**Author Contributions:** Conceptualization, X.S., T.M. (Takanori Matsui) and T.M. (Takashi Machimura); Formal analysis, X.S., T.M. (Takanori Matsui) and T.M. (Takashi Machimura); Methodology, X.S., T.M. (Takanori Matsui) and T.M. (Takashi Machimura); Supervision, T.M. (Takanori Matsui) and T.M. (Takashi Machimura); Writing—original draft, X.S.; Writing—review & editing, T.M. (Takanori Matsui), T.M. (Takashi Machimura), X.G. and A.H. All authors have read and agreed to the published version of the manuscript.

**Funding:** This research received no external funding.

**Acknowledgments:** We greatly appreciate the positive and constructive comments from the editors and the reviewers.

**Conflicts of Interest:** The authors declare no conflict of interest.

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
