# Peer review of "Toward Sustainable Development: Decoupling the High Ecological Footprint from Human Society Development: A Case Study of Hong Kong"

_sustainability, doi:10.3390/su12104177_

Round 1
Reviewer 1 Report
The paper is well written and very thorughly and comprehensively referenced.
One issue that I have with not only this paper but with almost all papers on sustainability is the use of wording that infers that with some changes in energy use, transport, etc., a city, region, state or country can become 'sustainable'. For examp,e, in line 18 in the Abstract there is a sentance the aim of the paper is to make policy recommendations to make HK a sustainable city. In line 278 there is a statement that Singapore was transformed into a sustainable city. The reality is that our cities and most regions, states and countries are so far from being sustainable that that is simply a dream. I am careful even with the term 'more sustainable' as what it really should say is 'less unsustainable'. All this is to say that those writing on the topic of sustainability need to think carefully about the terminology and what it means.
What this paper does is in fact to take some environmental, social and economic parameters and to show that (a) in the case of Hong Kong, there has been a deterioration in environmental parameters, and (b) vy comparing Hong Kong with Singapore, suggesting polcy meanures that could be taken to improve those environmental parameters.
Specific comments
Line 25: The focuws must be on ... protection. Protection of what?
Line 46 It can access should be "Ít can be used to assess ..."
Line 65. The HD ghas become widely accepted in the sustainable development fields should be "HDI has become widely accepted as a useful metric in the sustaiability field"
line 90. , the seafood consumption of HK" shoudl be "Seafood consumption in HK"
Line 93 "renewable energy only accounted fro" should be "renewable energy accounted for only ... "
Line 120. Should be "Due to its geographical location, half of the world's population can be ...."
Line 123. The total population in 2016 was 7.34 million
Line 126. the average population growth rate from 1995 to 2016 was 1.08% per annum ...
Line 136: "enjoys a high reputation worldwide". Th sis vague. A high reputation for what?
Table 1. LPG, natural gas, coal, electricity should follow on fom the line above.
Table 2. The gaseous fuels listed included LPG, Gas, Natual gas. What is 'Gas'? Is it biogas, compressed natural gas, ... ?
Line 224. There needs to be a space betwee m the equation in line 223 and line 224
Line 226. There needs to be a space betwee m the equation in line 225 and line 226.
Figure 4. Refers to 'each bar'. I could be incorrectr but I don't think that 'bar' is the correct term here as this is not a bar graph but a line graph with areas under the lines.
Line 247. Should be 'The most common types of fresh food ... were vegetables, ... '
Line 292. 'irresponsible' human activities. This is a bit emotive or value laden. Iwould suggest simply 'these human activities are eroding'
Section 4. Discussion and Implications. This is not so much a discussion as a new section that introduces Singapore andf makes comparisons with HK. I would be inclined to rename it ''HK and Singapore: A Comparitive Analysis
Line 340. The sustanable development of human society not only relates to the improvements of humanity but ...'should be 'Sustanable development of human society relates not only to the improvements of humanity, but ...
Line 365 although sustainability ...sustained ...' That does not read well. Perhaps it should be ' ... although sustainability ... was maiantained at a high level ...'
Line 416. 'the liquefied natural gas ... has been started importing' should be 'iquefied natural gas ... has started to be imported'
Lines 498 to 514. All of the 'Do' at the start of each policy recommendation are unecessary and can/should be deleted. For example, in line 498 'Accelerate the develpment'
Lines 524 - 540. Ditto!
Line 552. HK demonstates a high resource consumpion of citizens. should be 'HK is characteriesed by high percapita resource consumption'
Line 554. To citizens DO NOT overconsume. Such an appeal to the public to change behaviour without also suggesting how that could be best achieved is trite. It should be reworded as a policy statement along th eline sof Encourage members of the public to consume less using educational campaigns, which is esstentially what is suggested.
560 ' ... advertisment to overconsumption' should be 'not to overconsume'
Line 561. the DO can be omitted. 'Separate duties ...'.
This policy recommendation is in fact problematcal as there is no evidence provided anywhere in the paper that there is a lack f separatrion of responsibilities between government agencies r that thi sis the cause of a problem. So it is ne winformation dropped into a conclusion/recomendation.
Line 570. The paper combines analysis of ecological footprint ... should be 'This paper provides an analysis that combines ecological footprint, ... '
Line 578. 'to decouple'. This term was made ppular by Amory Lovins in discussing ways to change th erelationship between energy use and GDP. While it sounds impressive, it is in fact inaccurate, misleading and confusing to use the term 'decouple' as there will remain a relationship or link between the ecological footprint and human development, but th erelationship will change.
Line 590. 'prompted'. Should this be 'promoted'?
Line 597. The aim is coming up with' should be 'The aim is to come up with ... '
References.
General
There is inconsistency in terms of whether or not a period (full stop) is used at the end of each refernece. For example th efirst reference has a full stop at the end ( ... 2017.) while the second refernce does not ( 641-642)
There is no and betwen the second last and last autors cted. For example, reference 3 should be Wackernagel M, Hanscom L and Lin D. This comment pertains to many of the other refernences cited.
Reference 16. 'Prog. Geogr'. References should be cited using the full name of the journal
Refernce 18. 'Ying yong sheng tai xua bao='. It is unclear why this is in the cited reference
1. 'Master' should be 'Master thesis'
2. Sustainability science should be Sustainability Science
3.
Reviewer 2 Report
The article deals with the analysis of the ecological footprint, the human development index and the biocapacity of Hong Kong. The authors compare these results with those of Singapore and perform a SWOT analysis in order to provide recommendations for a sustainable development.
The manuscript is well done with a clear structure and the arguments are well exposed.
I suggest the authors to provide an estimation of the future trend of the indices if their recommendations were developed.
Reviewer 3 Report
This paper deals with a quite useful topic. It is quite well written.
The graphs in the paper should reference the data sources on which calculations in them are based.
Other suggestions for improvement of the paper are:
Lines 165 to 172. The sources of the equations discussed in these lines should be shown.
Line 171. The term "EQFi" and its significance should be discussed in more depth.
Line 208 to 210. The term "HDI" and its significance should be discussed in more detail.
Line 440. Figure 10 does not appear to be discussed in the paper.
Line 464. "lead" should be "led".
Line 470. "the SWOT" should be "a SWOT".
Lines 486 to 567. It is suggested that "DO" be replaced by "It is recommended that" or similar wording; and that "DO NOT" be replaced by "It is recommended that ..... do not", or similar wording.
Tables S1, S2 and S3 should be referenced.
